# Two cold shock domain containing proteins trigger the development of infectious *Trypanosoma brucei*

**Justin Y. Toh[1]ʘ, Agathe Nkouawa[1]ʘ, Gang Dong[2], Nikolay G. Kolev[1], Christian Tschudi[1]***

**1** Department of Epidemiology of Microbial Diseases, Yale School of Public Health, New Haven, Connecticut, United States of America, **2** Max Perutz Labs, Vienna Biocenter, Center for Medical Biochemistry, Medical University of Vienna, Vienna, Austria

ʘ These authors contributed equally to this work.
* christian.tschudi@yale.edu

**Data Availability Statement:** RNA-Seq data from this study have been submitted to the NCBI Sequence Read Archive-SRA at http://www.ncbi.nlm.nih.gov/Traces/sra/sra.cgi with accession

## Abstract

Cold shock proteins are members of a family of DNA- and RNA-binding proteins with one or more evolutionarily conserved cold shock domain (CSD). These proteins have a wide variety of biological functions, including DNA-damage repair, mRNA stability, and regulation of transcription, splicing and translation. We previously identified two CSD containing proteins, CSD1 and CSD2, in the protozoan parasite *Trypanosoma brucei* to be required for RBP6-driven metacyclic production, albeit at different steps of the developmental program. During metacyclogenesis *T. brucei* undergoes major morphological and metabolic changes that culminate in the establishment of quiescent metacyclic parasites and the acquisition of mammalian infectivity. To investigate the specific role of CSD1 and CSD2 in this process, we ectopically expressed CSD1 or CSD2 in non-infectious procyclic parasites and discovered that each protein is sufficient to produce infectious metacyclic parasites in 24 hours. Domain truncation assays determined that the N-terminal domain, but not the C-terminal domain, of CSD1 and CSD2 was required for metacyclic development. Furthermore, conserved amino acid residues in the CSD of CSD1 and CSD2, known to be important for binding nucleic acids, were found to be necessary for metacyclic production. Using single-end enhanced crosslinking and immunoprecipitation (seCLIP) we identified the specific binding motif of CSD1 and CSD2 as "ANACAU" and the bound mRNAs were enriched for biological processes, including lipid metabolism, microtubule-based movement and nucleocytoplasmic transport that are likely involved in the transition to bloodstream form-like cells.

## Author summary

*Trypanosoma brucei* is a protozoan parasite that causes African sleeping sickness in humans and together with *T. congolense* and *T. vivax* remains a public health concern in sub-Saharan Africa by afflicting livestock with nagana. During its complex life cycle between the mammalian host and the blood-feeding tsetse fly vector (*Glossina sp.*), *T. brucei* relies on several differentiation steps, where parasites transition between replicative and non-replicative forms specialized for infectivity and survival in mammalian and tsetse

numbers: PRJNA898598, PRJNA898614, PRJNA899536 and PRJNA899539.

**Funding:** This work was supported by National Institutes of Health (http://www.nih.gov) grants AI028798, AI110325, AI165480 and AI007404 to CT. JYT was supported in part by a training grant from the National Institute of Allergy and Infectious Diseases of the National Institutes of Health (T32 AI007404 to CT). The funders had no role in study design, data collection and analysis, decision to publish, or preparation of the manuscript.

**Competing interests:** The authors have declared that no competing interests exist.

fly hosts. The transmission cycle begins with the tsetse fly taking a bloodmeal from an infected mammal. Following establishment of non-infective midgut procyclics, trypanosomes must find their way to the salivary glands where they develop into quiescent and infective metacyclics. We previously identified two cold shock domain containing proteins, CSD1 and CSD2, to be required for this developmental process. In this study, we determined that each protein is sufficient to trigger the development of infectious metacyclic parasites in 24 hours. The N-terminal domain of both proteins and conserved amino acids in the cold shock domain known to bind amino acids were required for this process. In addition, CSD1 and CSD2 bind mRNAs through the sequence ANACAU and preferentially target mRNAs that help to establish parasites capable of surviving in the mammalian host.

## Introduction

*Trypanosoma brucei* is a protozoan parasite and the etiologic agent of African sleeping sickness, a major public health concern in Sub-Saharan Africa. *T. brucei* is transmitted to humans and other mammals by the bite of an infected tsetse fly (genus *Glossina*). Consequently, these parasites have evolved specific adaptations to allow for their survival in both the gut and salivary glands of the tsetse fly, as well as in their mammalian hosts. The proliferative slender form parasites that is responsible for causing the disease is compartmentalized in the blood, adipose tissue, skin, and, in the later stages of the infection, in the central nervous system. The persistence of the slender form parasites in the mammalian host is largely due to their variant surface glycoprotein (VSG) coating, which enables immune system evasion *via* antigenic variation. Once slender form parasites propagate to a high cell density, a quorum sensing response triggers the transition to quiescent stumpy form parasites, which are primed for survival in the tsetse fly [1,2]. Within the tsetse fly, procyclic parasites undergo metacyclogenesis [3,4], a complex developmental process whereby noninfectious trypanosomes develop into metacyclic parasites that are infectious to mammals due to the expression of a subset of VSGs known as metacyclic VSGs (mVSGs) [5]. Procyclic parasites are initially found in the midgut, where they shed the VSG coat and instead express a glycoprotein coat composed of procyclins that are rich in either Gly-Pro-Glu-Glu-Thr repeats (GPEET) or Glu-Pro repeats (EP) and display elaborate mitochondrial cisternae for an oxidative phosphorylation-dependent ATP production and metabolism that uses proline as the primary carbon source [6–9]. As procyclic parasites transit to the foregut/midgut junction known as the proventriculus, they elongate and reposition their kinetoplast (mitochondrial genome) anterior to the nucleus to become epimastigote parasites, which undergo asymmetric division to produce long and short epimastigote daughter cells. Although long epimastigotes likely degenerate, short epimastigotes that travel to the salivary gland express a family of alanine-rich surface proteins known as brucei alanine rich protein (BARP) and colonize the salivary glands by attaching to the epithelium [10]. A subset of the attached epimastigotes will undergo asymmetric division to produce nonattached, infectious, quiescent metacyclic parasite that are primed for survival in the mammalian host [11]. Metacyclic parasites characteristically display a regressed mitochondrion in preparation for a glycolysis-based ATP generation, reposition their kinetoplast to the posterior pole of the parasite body, increase endocytosis, and express mVSGs for survival in the mammalian host [12].

Although the major morphological changes during metacyclogenesis have been well-documented [3,4], the molecular mechanisms that drive this developmental process remain largely unknown. We have previously established an *in vitro* differentiation system by overexpressing

the single RNA binding protein 6 (RBP6) in procyclic parasites [12]. This system recapitulates most of the developmental events leading to metacyclic parasites and has significantly assisted research into the underlying biological processes that occur during metacyclogenesis by enabling large-scale genetic and biochemical experiments. Multi-omics profiling studies have resulted in the molecular documentation of the transcriptomic, proteomic, and metabolomic changes in procyclic parasites that develop into epimastigote and metacyclic parasites [13–15]. In particular, transcriptomic and proteomic analysis of metacyclics displayed a dramatic reduction in general transcription and protein synthesis, which is consistent with the quiescent, non-dividing state of these parasites [13]. Additionally, there was a corresponding up-regulation of glycolytic proteins at both the transcript and protein level, which supports the metabolic priming of metacyclic parasites towards glycolysis as the primary method of ATP production. The information provided by the transcriptomic and proteomic studies were also the basis for a targeted RNAi screen to identify essential genes for RBP6-driven metacyclogenesis, which identified 22 genes involved in metacyclic production [16]. In the study presented here, we focused on two genes that were essential for metacyclic development, namely the cold shock domain containing protein 1 (CSD1; Tb927.8.7820) and CSD2 (Tb927.4.4520).

The cold shock domain (CSD) is a nucleic acid binding domain that is found in many prokaryotic and eukaryotic organisms. This domain was first discovered to function in cold acclimation of *Escherichia coli*, as cold shock proteins A (CspA), CspB, CspG, and CspI were inducible under low temperature to act as RNA chaperones to destabilize RNA secondary structures that develop at low temperatures [17–20]. However, it was soon realized that many cold shock domain containing proteins were not cold shock inducible, functioned outside of cold acclimation, and were involved in housekeeping functions. For instance, CspC and CspE are expressed constitutively and involved in chromosome partitioning [21], while CspD is induced upon entry into stationary phase and upon carbon starvation, while acting as a replication inhibitor by binding to ssDNA [22]. In mammalian cells, the most widely studied cold shock domain containing protein is the Y-box binding protein 1 (YB-1), which binds both DNA and RNA and interacts with many proteins to regulate transcription, translation, drug resistance, cell proliferation, early embryonic development, differentiation and stress adaptation [23,24]. The *T. brucei* genome encodes a total of four cold shock domain containing proteins, namely CSD1, CSD2, CSD3 (Tb927.7.3810), and RNA binding protein 16 (RBP16, Tb927.11.7900). Whereas CSD3 has not yet been characterized, RBP16 is a mitochondrial Y-box family protein essential for RNA editing [25–27]. CSD1 and CSD2 were originally identified as being required for RBP6-driven metacyclic production [16], but their specific role in this process remains unknown. Through the ectopic expression of CSD1 or CSD2 in procyclic parasites, we discovered that each protein is sufficient to produce metacyclic parasites in 24 hours. Our inducible expression system provided a quick readout to functionally dissect the protein domains of CSD1 and CSD2. Furthermore, we were able to identify the mRNA targets, as well as the specific binding motif of CSD1 and CSD2 using single-end enhanced crosslinking and immunoprecipitation (seCLIP).

## Results

### Overexpression of CSD1 or CSD2 in procyclic parasites causes rapid generation of metacyclic cells

Prior knockdown by RNAi of CSD1 or CSD2 in the background of RBP6-driven metacyclic production exhibited very strong negative effects on this process, albeit at different steps of development [16]. CSD2 RNAi negatively affected the repositioning of the kinetoplast and morphological appearance of epimastigotes [16]. In contrast, CSD1 RNAi displayed normal kinetoplast repositioning, but brucei alanine-rich protein (BARP) expression, a diagnostic

marker for short epimastigotes, was decreased by approximately 80% [16]. To further characterize the role of CSD1 and CSD2, we separately overexpressed the two genes in procyclic trypanosomes (*T. brucei* Lister 427 29–13 strain) and first examined the cultures microscopically. Quite unexpectedly, within 24 h of CSD1 or CSD2 induction we observed that 40–50% of the cells appeared morphologically as metacyclics with an undulating membrane and bloodstream form (BSF)-like corkscrew motility (Fig 1A and 1B). In contrast, overexpression of CSD3 and RBP16 did not result in production of metacyclics, but cells with an epimastigote-like morphology were detected in these cultures (S1 Fig). Additionally, both the CSD1 and CSD2 cell lines displayed an arrest in proliferation and the metacyclics appeared to be non-dividing (Fig 1C). These observations strongly suggested that CSD1 and CSD2 overexpression resulted in

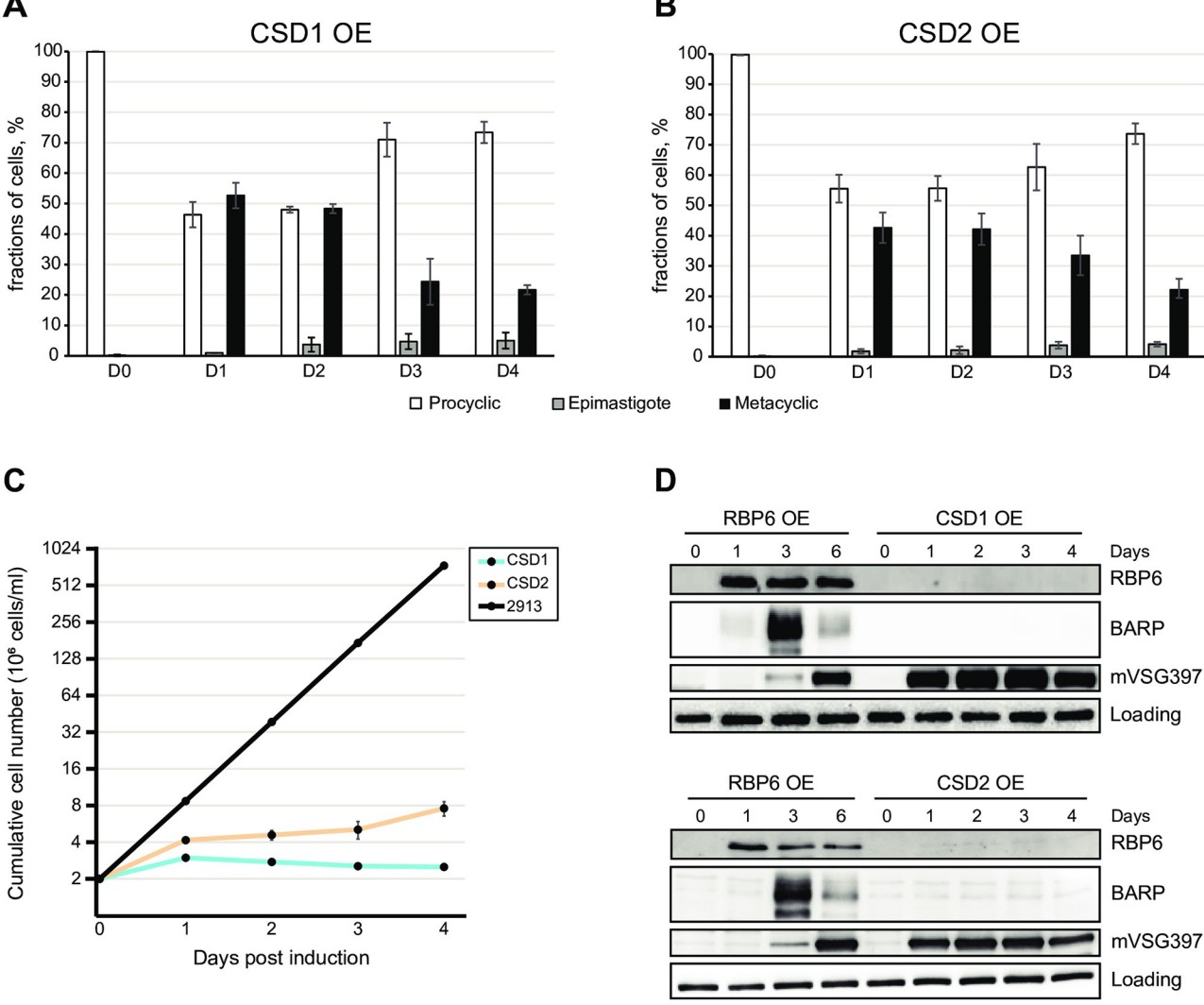

**Fig 1. CSD1 and CSD2 overexpression triggers metacyclic development. (A,B)**. The induced CSD1 and CSD2 overexpression (OE) cell lines were monitored daily by light microscopy up to 4 days and the presence of various developmental trypanosome forms was based on morphology after DNA staining [12]. Three independent biological replicates were performed and at least 100 cells were counted for each time point with means ± standard deviation (std). **(C)**. Cell numbers were determined every day for 4 days and plotted logarithmically. Three independent replicates were performed and at least 100 cells were counted for each time point with means ± standard deviation (std). **(D)** Aliquots were processed for Western blot analysis to quantify BARP and mVSG397 expression and compared to the positive control, the wild-type RBP6 overexpression cell line (RBP6 OE). Elongation factor 1-alpha (EF-1α) was used as a loading control as described [16].

rapid metacyclic production. To verify this conclusion, we monitored the CSD1 and CSD2 inducible expression cell lines by Western blotting for the expression of the metacyclic marker mVSG397. After a 24 h induction, both cell lines displayed mVSG397 protein expression levels comparable to that of the 6-day induced RBP6 cell line (Fig 1D). Visual inspection by microscopy of the CSD1- and CSD2-induced cultures at the 4, 8, 16, and 24 h timepoints indicated that the intermediate epimastigote stage was skipped. To confirm this observation, we monitored the levels of BARP, a diagnostic marker for short epimastigotes by Western blot analysis. As expected, in cultures where RBP6 was induced the levels of BARP protein were highly up-regulated, as compared to un-induced cells (Fig 1D). In contrast, induction of either CSD1 or CSD2 revealed no detectable BARP protein (Fig 1D) and overexpression of CSD3 and RBP16 did not result in detectable BARP or mVSG397 protein (S1 Fig). Thus, CSD1 and CSD2 over-expression induced a developmental program generating metacyclic trypanosomes in 24 h without a transition to the intermediate epimastigote stage.

Since we noted that metacyclics generated by overexpression of CSD1 or CSD2 were not dividing, we determined their position in the cell cycle by scoring the number of nuclei and kinetoplasts. Metacyclic parasites in the RBP6 overexpression system are arrested in the G1/G0 phase of the cell cycle with over 95% of the cells displaying a one kinetoplast and one nucleus (1K1N) configuration [12]. In contrast, the overexpression of RBP6 with a single point mutation (Q109K) generated metacyclic parasites arrested outside of the G1/G0 checkpoint, with 75% of the cells displaying the 1K1N configuration and 2K1N and 2K2N cells representing 6% and 13% of the metacyclic population, respectively [28]. Purified metacyclics [13] from the CSD2 overexpression cell line quantitatively accumulated in the 1K1N configuration, representing mainly G1/G0 cells (Fig 2A). In addition, there was no evidence for kinetoplast elongation in 1K1N cells, excluding the possibility that they have entered S phase [29]. In contrast, 77% of purified metacyclics [13] from the CSD1 overexpression cell line were in the 1K1N

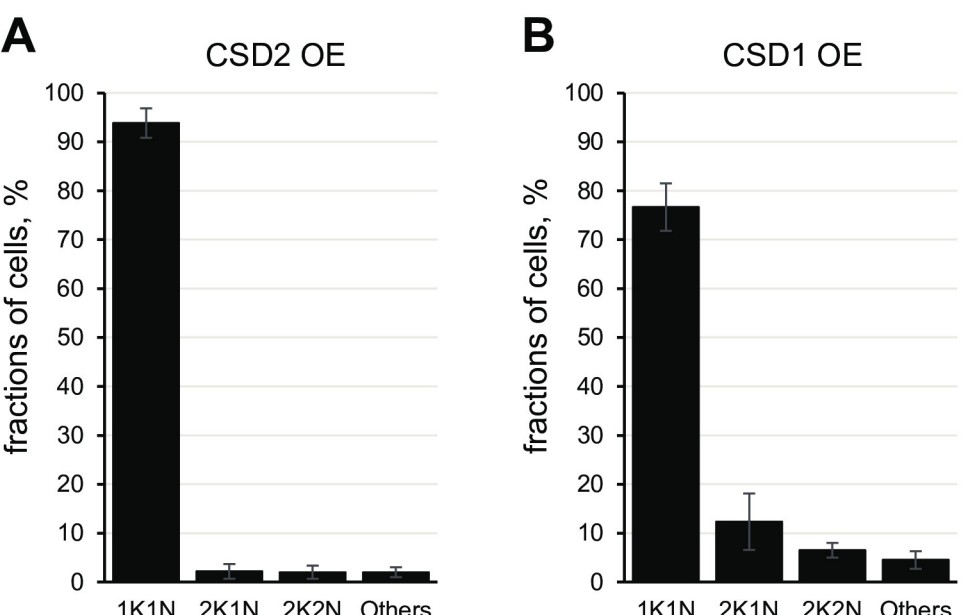

**Fig 2. Karyotype of metacyclics following induction of CSD1 or CSD2.** Purified metacyclics from CSD2 (**A**) or CSD1 (**B**) OE cell lines were scored for the presence of 1 kinetoplast and 1 nucleus (1K1N), 2 kinetoplasts and 1 nucleus (2K1N), 2 kinetoplasts and 2 nuclei (2K2N) or other configuration (Others). Three independent biological replicates were performed and at least 100 cells were counted with means ± standard deviation (std).

configuration, 12% were 2K1N, 6.5% displayed the 2K2N cell configuration, and 4.5% were multinucleated or displayed more than 2 kinetoplasts (Fig 2B). This distribution is similar to that of the metacyclics generated from the RBP6 mutant [28], indicating that a significant population of CSD1 produced metacyclic parasites arrested outside of the G1/G0 checkpoint, which could suggest defects in kinetoplast replication and segregation and/or cytokinesis.

## Domain requirements for CSD1 and CSD2 function

CSD1 and CSD2 have a relatively low overall sequence conservation with other cold shock domain proteins, including YB-1 and Lin28. Nevertheless, a central stretch of ~110 amino acids revealed significant homology (~30% identity and ~40% similarity) to other cold shock domains (CSD) and a number of key residues important for CSD function are conserved in the *T. brucei* proteins (Fig 3A). The CSD is a well conserved nucleic acid binding domain that is homologous to the bacterial cold shock proteins and adopts a characteristic oligonucleotide/oligosaccharide-binding (OB) fold, consisting of five anti-parallel β-strands to form a β-barrel [30,31]. Compared with their bacterial homologs and YB-1, one noteworthy structural difference in the CSD of CSD1 and CSD2 is that an extended loop follows the β2-strand by 40 and 41 amino acids, respectively (Fig 3A). This loop is overall negatively charged (pI ~4), which is in contrast to the positive charge of the folded region of the CSDs (pI ~10). The CSD of CSD1 and CSD2 are flanked by a proline- and serine/threonine-rich N-terminal domain (NTD) of ~210 amino acids and a C-terminal domain (CTD) of ~180 amino acids with many negative charges, i.e. D and E with an overall pI ~4 (Fig 3B). Like many other eukaryotic cold shock domain containing proteins, the NTD and CTD of CSD1 and CSD2 are bioinformatically predicted by the Russell/Linding propensity scale to be intrinsically disordered [32]. To determine whether the NTD and the CTD of CSD1 and CSD2 were required for inducing metacyclic production, we deleted either the NTD, the CTD or both the NTD and CTD. These deletions were introduced in CSD1 and CSD2 versions containing a 3xFLAG epitope at the C-terminus, which did not affect the production of metacyclic parasites. All cell lines expressed the truncated proteins at levels similar or higher to that of the wild-type overexpression cell line (Fig 4). Deletion of the CTD in CSD1 or CSD2 induced metacyclic production to a level comparable to wild-type cell lines, as indicated by the expression of mVSG397 (Fig 4A and 4B). However, deletion of either the NTD alone or both the NTD and CTD in CSD1 or CSD2 did not result in the accumulation of mVSG397 (Fig 4A and 4B). This suggested that the NTD plays an important role in driving metacyclic production. We used several bioinformatic tools to further determine, if there were any specific features in the NTD of CSD1 and CSD2 to fulfill its function in metacyclic production. The Prion-Like Amino Acid Composition (PLAAC) program that uses a hidden-Markov model algorithm to identify probable prion subsequences [33] predicted a prion-like domain in the NTD of both CSD1 and CSD2 (S2 Fig). However, deletion of this predicted prion-like domain did not affect metacyclic production (S3 Fig) and thus, further experiments will be required to determine why the NTD is required in driving metacyclic production.

The extended loop following the β2-strand in the CSD of CSD1 and CSD2 has a central stretch of 32 and 33 residues, respectively, with little sequence conservation (highlighted in green in Fig 3A). Whereas deletion of the 32 residues in CSD1 did produce metacyclics (Fig 5A), deletion of the 33 residues in CSD2 did not result in the accumulation of mVSG397 (Fig 5B). Since the deletion in CSD2 is 1 amino acid longer, we decided to add 1 amino acid (Glu or Ala) back (Fig 5C), thus making the deletion 32 residues, as in CSD1. Both Glu or Ala addition rescued the function of the protein, suggesting some spatial constraints in the 33 residue deletion, which might affect the proper folding of the CSD.

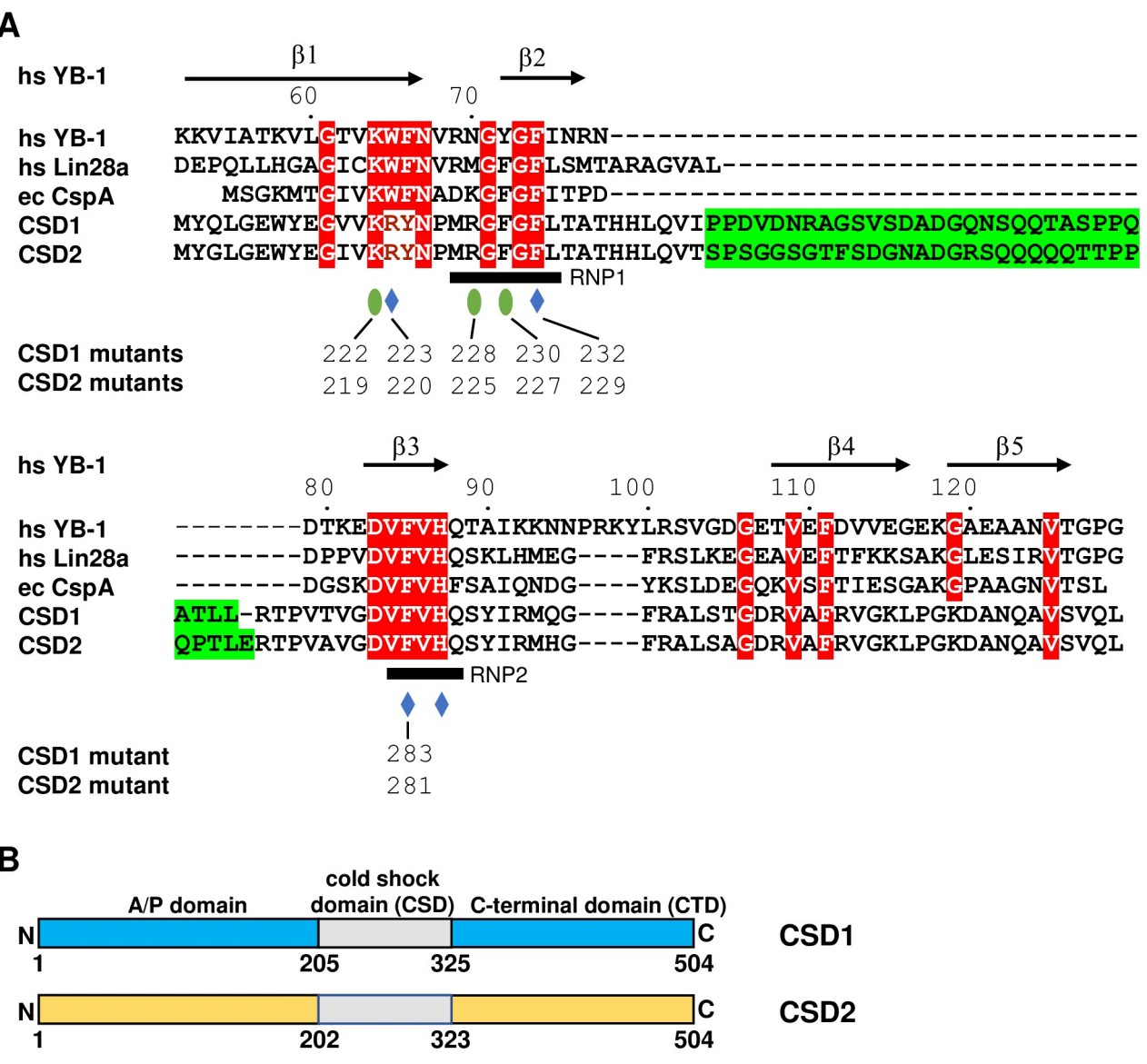

**Fig 3. Structure based-multiple sequence alignment of the CSD1 and CSD2 cold shock domain with other eukaryotic and bacterial CSDs and structural overview of CSD1 and CSD2.** (A) Sequence alignment of the cold shock domain of human YB-1 (hs YB-1), human Lin28a (hs Lin28a), *Escherichia coli* cold shock protein A (ec CspA), CSD1 and CSD2. The secondary structure features of hs YB-1 CSD are indicated above the sequences [30] and RNP1 and RNP2 are highlighted by a black bar. Identical residues are indicated as white letters with red shading. Blue diamonds, aromatic residues involved in π-π stacking interactions. Green ovals, important residues in YB-1 facilitating RNA interactions [30]. The central loop, as highlighted with green shaded residues, encompasses non-conserved residues in the loop of CSD1 and CSD2. (B) CSD1 and CSD2 are organized into three domains: an alanine and proline-rich N-terminal domain, a central cold shock domain (CSD), and a C-terminal domain (CTD).

Next, we decided to test whether the domains were functionally interchangeable. Therefore, we created 8 chimeric constructs where the NTD, CTD, NTD and CTD, or the non-conserved sequence in the CSD loop was swapped between CSD1 and CSD2 (S4 Fig). Each of the 8 constructs was individually overexpressed in the *T. brucei* Lister 427 (29–13) procyclic cell line and found to display wild-type levels of mVSG397 expression at the 24 h and 48 h timepoint. Thus, all the domains tested were interchangeable and the chimeras caused rapid metacyclic development.

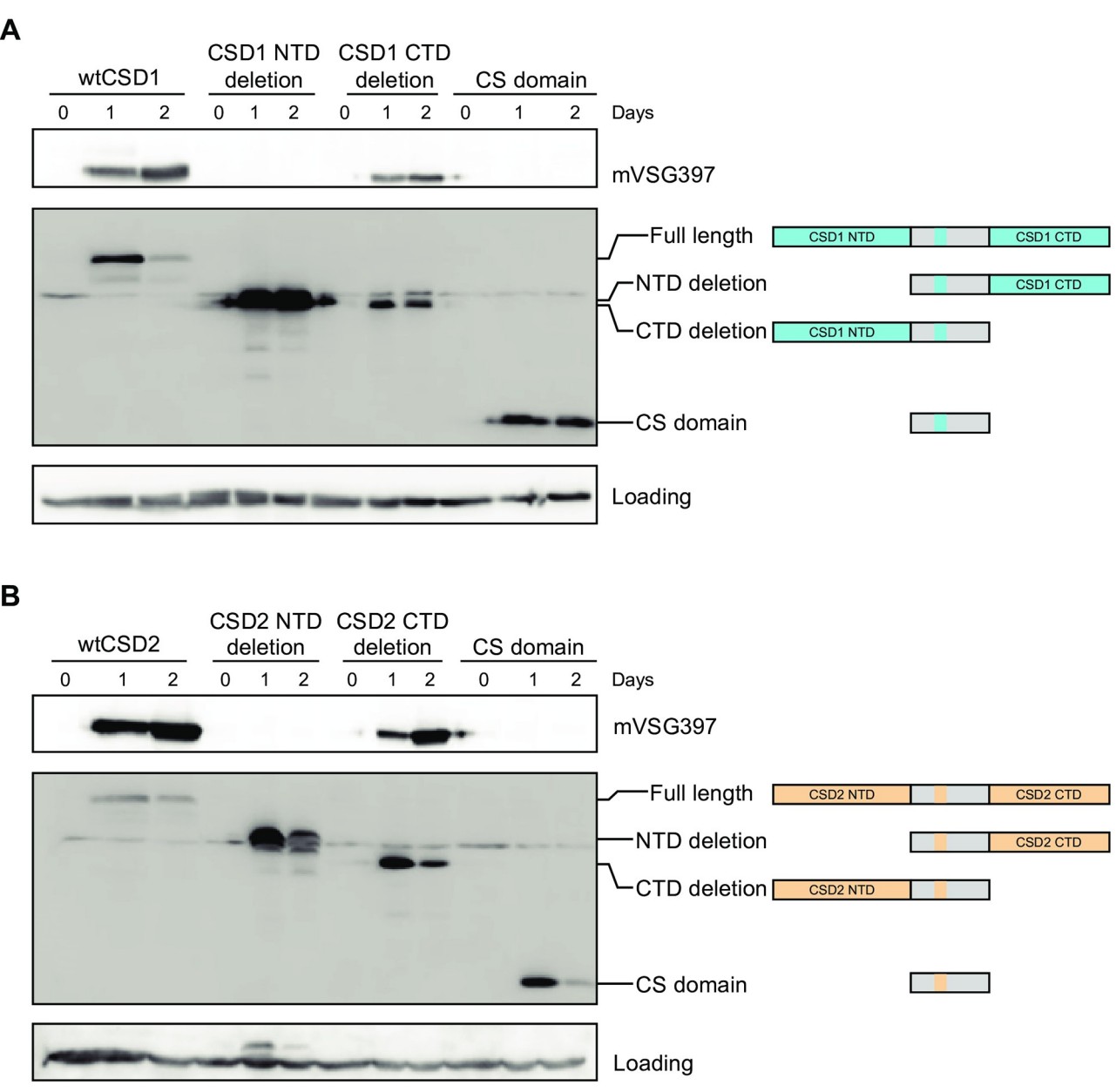

**Fig 4. The NTD is essential for inducing mVSG expression in both CSD1 and CSD2.** Procyclic parasites expressing a truncated version of CSD1 (**A**) or CSD2 (**B**) as indicated on the right, were assayed by Western blot for mVSG397 and the truncated protein. wtCSD1 or wtCSD2 were used as positive controls.

With the aid of homology-based structural modelling and the existing detailed information about important functional residues in the CSD, which includes the crystal structure of the Lin28 CSD in the absence and presence of nucleic acids [34] and the crystal structures of the YB-1 CSD in complex with different RNA oligos [30], we predicted key residues in the CSD1 and CSD2 CSD (S5 Fig), targeted them by mutagenesis and then assayed their effect on *in vitro* metacyclogenesis. Like prokaryotic and eukaryotic CSDs, the CSD1 and CSD2 CSD contain two consensus motifs, RNP1 and RNP2, in the predicted β2-strand and β3-strand, respectively. These motifs have three highly conserved residues (F74, F85 and H87 in YB-1 in Fig

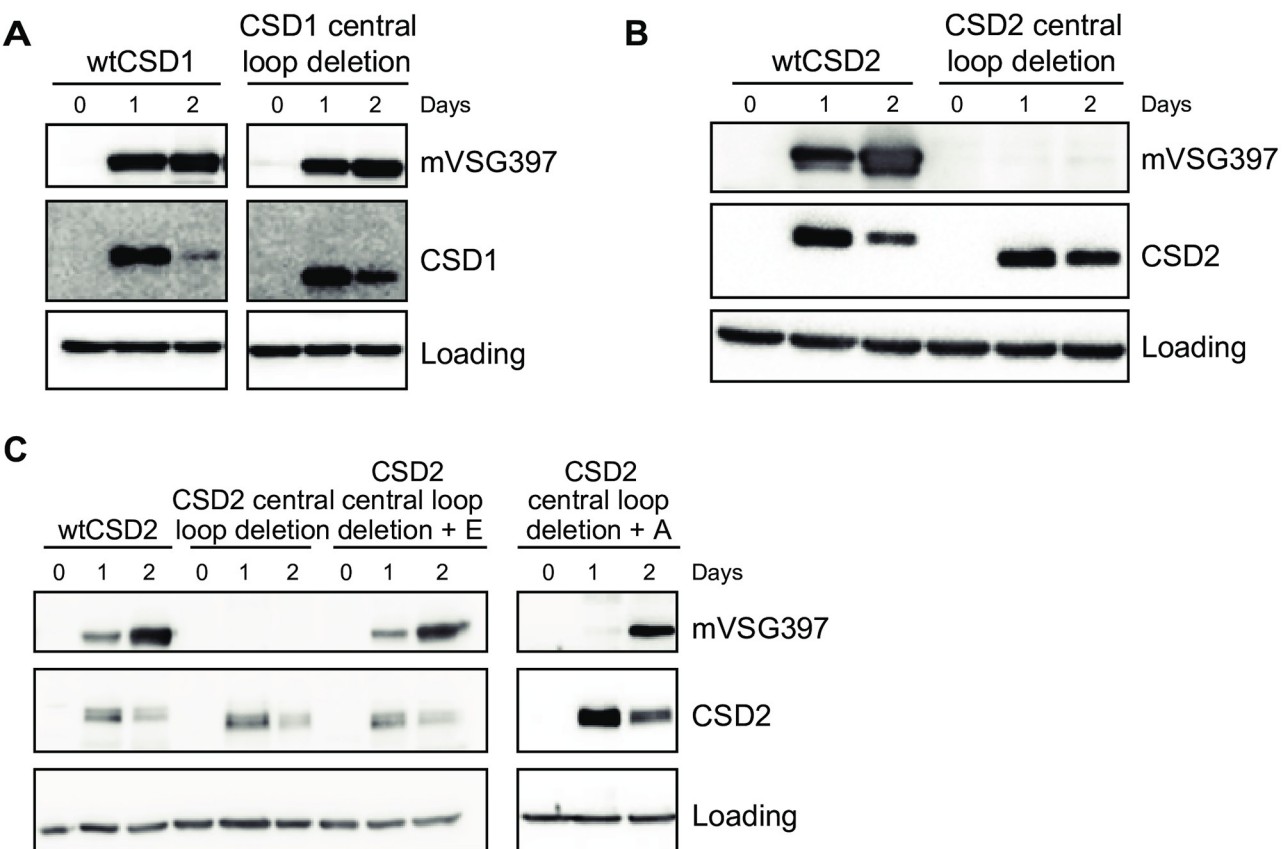

**Fig 5. The central loop present in the cold shock domain in CSD1 and CSD2 is not essential for inducing mVSG expression.** The central loop sequence as highlighted in green in Fig 3 was deleted in CSD1 (**A**) and CSD2 (**B**). Addition of either Glu or Ala to the CSD2 deletion restored metacyclic production (**C**).

3A), which form a hydrophobic surface patch with W65 on YB-1 (Figs 3A and S5) in the β1-strand and they are key residues interacting with RNA through π-π stacking. In addition, there are several other important residues in YB-1 facilitating RNA interactions, including K64, N70 and Y72 (Fig 3A). To assess the functional significance of the corresponding residues in CSD1 and CSD2, we generated a triple mutation by changing two residues in RNP1 and one residue in RNP2 (Fig 3A) and a double mutation in RNP1 (Fig 3A). In addition, we mutated three residues corresponding to K64, W65 and N70 in YB-1 and also tested a double mutant changing the residues corresponding to K64, W65 (Figs 3A and S5).

We overexpressed the mutant 3xFLAG-tagged genes in procyclic parasites and the levels of the mutant proteins were examined by Western blot with anti-FLAG antibodies. The abundance of all eight mutant proteins was approximately 100-fold lower relative to the wild-type protein at the 24 h timepoint, which could be due to less expression of these mutant proteins or to protein instability. Therefore, to equalize protein levels between the positive control and the mutants, we induced the wild-type CSD1 (wtCSD1) and wtCSD2 cell lines with 1,000-fold less doxycycline (10 ng/ml). These induction conditions equalized protein expression levels, but delayed maximum mVSG397 expression to the 48 h timepoint (Fig 6). Under these modified conditions overexpression of the eight mutated proteins did not yield any detectable levels of mVSG397 at the 48 h timepoint, confirming that the CSD in CSD1 and CSD2 plays an essential functional role, likely involving nucleic acid binding (Fig 6).

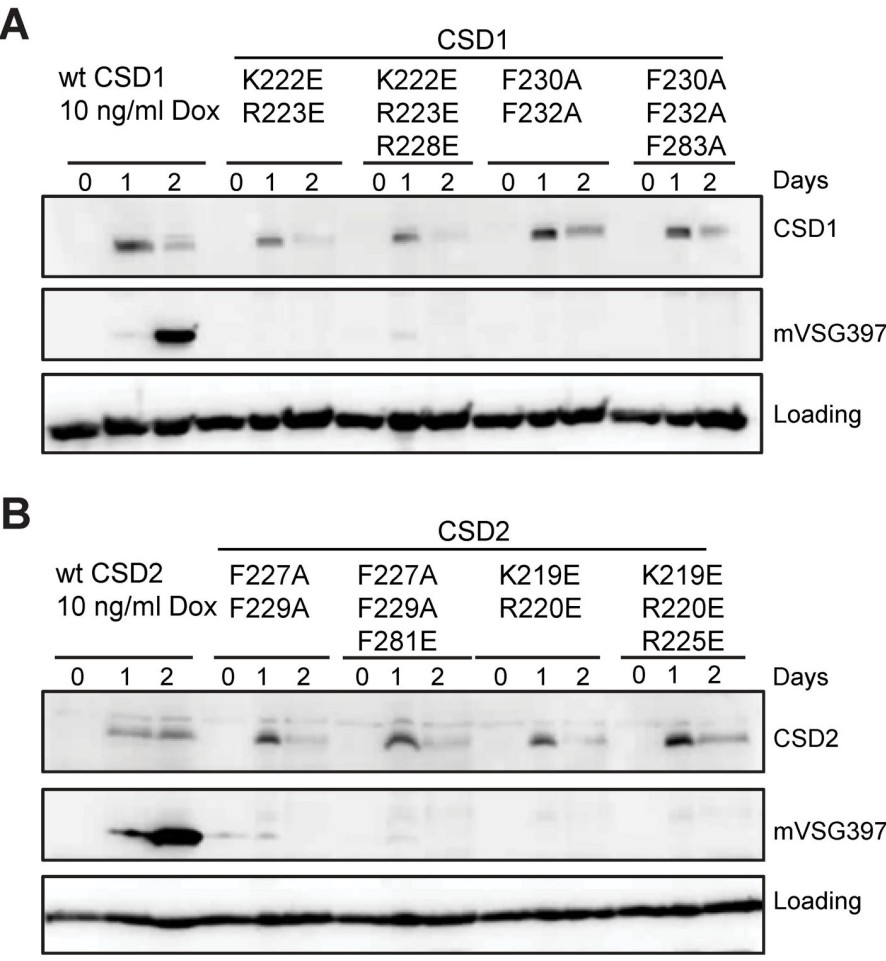

**Fig 6. Conserved amino acids in the CSD1 and CSD2 cold shock domain, known to bind nucleic acids in YB-1 and Lin28, are required to induce mVSG expression.** Conserved amino acids as highlighted in Fig 3 were mutated in CSD1 (**A**) and CSD 2 (**B**) as indicated above the panel. Western blots were performed to monitor the expression of the mutated protein and mVSG397. The positive control wtCSD1 and wtCSD2 was induced with 10ng/ml doxycycline, which is 1,000-fold less than what is normally used, to normalize protein expression levels between the controls and the experimental samples.

## Differential transcript abundance during the CSD1- and CSD2-driven development from procyclic to metacyclic cells

To begin to understand events triggered by CSD1 and CSD2 in the rapid metacyclic production, we performed RNA-Seq on the CSD1 and CSD2 overexpression cell lines at the 1, 4, 16 and 24 h timepoints and compared the combined transcriptomes to the transcript abundance changes occurring during 6 days of RBP6 induction [28]. Inducible expression of CSD1 changed abundance levels of 564 transcripts over 24 h, with 322 and 242 transcripts up- and down-regulated, respectively (S1 Table). 185 (57%) of the 322 up-regulated transcripts were also observed in the background of RBP6 induction (Fig 7A and S1 Table) and included known regulators of metacyclogenesis [16]. Of the 137 transcripts up-regulated only in the CSD1 overexpression cell line, 54 (39%) were previously shown to be up-regulated >2-fold in BSF cells as compared to procyclic form (PF) cells [35]. In contrast, only 41 (17%) of the 242 down-regulated transcripts were present in the RBP6 induction (Fig 7A and S1 Table).

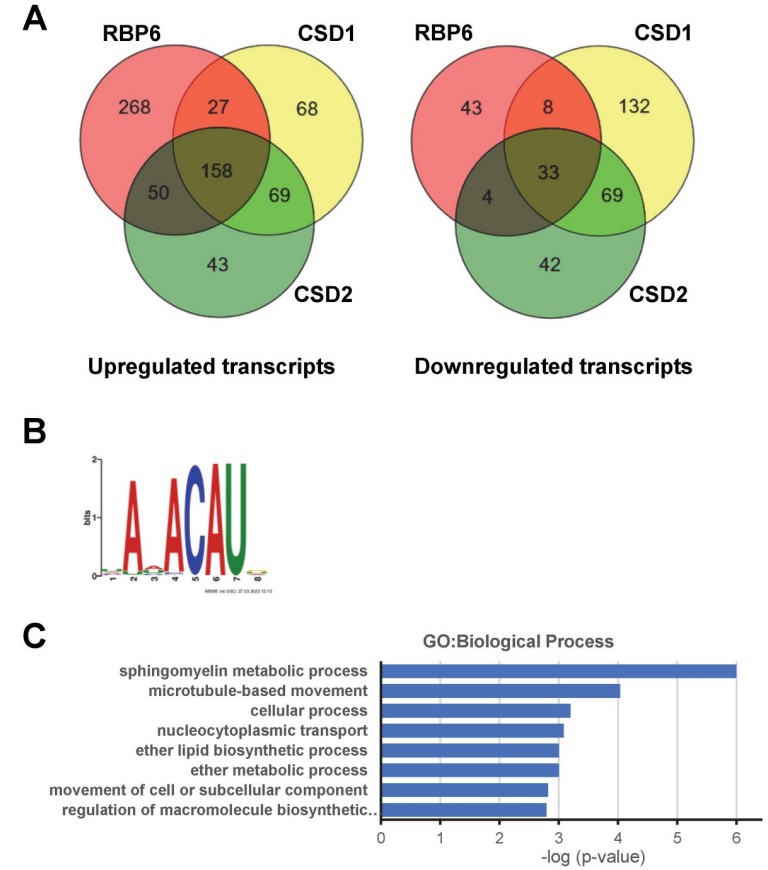

**Fig 7. Identification of CSD1 target transcripts. (A)** Overlap of differentially expressed transcripts in RBP6, CSD1 and CSD2 overexpressing cell lines. **(B)** CSD1 binding motif as determined by MEME [40]. **(C)** Functional annotation using the Gene Ontology (GO) enrichment tool on the Tri-TrypDB webserver (http://tritrypdb.org/) and GO terms were condensed by submitting to REVIGO [59].

Analyzing the transcripts altered more than two-fold by CSD2 overexpression revealed a total of 320 and 148 up- and down-regulated, respectively (S2 Table). As with CSD1, many of the up-regulated transcripts were in common with the RBP6 induction (208, 65%; Fig 7A and S2 Table), and this included the five mVSGs and known regulators of metacyclogenesis [16]. Interestingly, there was a very slight up-regulation of RBP6 (Tb927.3.2930) and BARP (Tb927.9.15530) transcripts with a 2.7- and 2.6-fold change, respectively (S2 Table). Similar to the CSD1 data set, only 37 (25%) of the 148 down-regulated transcripts were present in the RBP6 induction (Fig 7A and S2 Table). Taken together, the RNA sequencing showed that either CSD1 or CSD2 overexpression mimicked many of the changes induced by RBP6. In addition, we noted transcript changes in common for CSD1 and CSD2, but not occurring in the RBP6 cell line, namely 69 transcripts in both the up- and down-regulated cohort. In particular, the down-regulated transcripts were highly enriched for integral membrane proteins and procyclins (EP-2, EP3-2 and PARP A). Conversely, 5 epimastigotes markers, previously identified by single cell RNA-Seq of trypanosome development in the tsetse fly vector [36], were up-regulated around day 3 in the RBP6 induction, but did not significantly change in transcript abundance in the CSD1 or CSD2 overexpression (S6 Fig), providing further evidence that CSD1/2 overexpression did not result in the production of epimastigotes. Finally, there was a

small subset of transcripts that responded to overexpression of one RBP, but not the other. This suggested that there were some transcript changes unique to each RBP and confirmed our earlier results that CSD1 and CSD2 act in distinct steps of metacyclogenesis [16]. The 68 up-regulated transcripts specific to CSD1 included RBP7A (Tb92710.12090), a component of the stumpy induction factor (SIF) signaling pathway [2], PUF2 (Tb927.10.12660), and a serine/threonine-protein phosphatase 1 regulatory subunit (Tb927.10.11960), which is part of a complex involved in transcription termination by RNA polymerase II. Notable up-regulated CSD2-specific transcripts (43 total) included metacaspase (MCA1, Tb927.11.3220), a gene necessary for RBP6-driven metacyclic production [16], and RDK1 (Tb927.11.14070), which has been shown to inhibit stumpy to procyclic differentiation [37]. Finally, down-regulation of PUF9 (Tb927.1.2600) only in the CSD1 dataset might explain why CSD1-generated metacyclics arrested outside of the G1/G0 phase, since it was previously shown that RNAi depletion of PUF9 in procyclic parasites caused an accumulation of cells in the G2/M phase [38].

## RNA-Seq analysis of metacyclics

Since the morphological analysis by microscopy revealed that metacyclics generated in CSD1 and CSD2 overexpression cell lines accumulated at different stages of the cell cycle, we next interrogated the transcriptomes of purified metacyclics. This revealed 925 differentially expressed transcripts in CSD1 metacyclics when compared to un-induced procyclic trypanosomes, with 424 and 501 transcripts up- and down-regulated, respectively (S1 Table). CSD2 metacyclics had 1,290 differentially expressed transcripts when compared to un-induced procyclic trypanosomes, with 748 and 542 transcripts up- and down-regulated, respectively (S2 Table). Similar to our previous transcriptome analysis of the RBP6 induction system [13,28], metacyclics derived from inducible expression of CSD1 and CSD2 had a predominantly BSF-like transcriptome with the largest changes occurring in genes encoding cell surface components. Thus, the majority of changes for CSD1 (679, 73%) and CSD2 (909, 70%) were in common with what we described in RBP6 metacyclics [28]. Nonetheless, functional categorization by Gene Ontology (GO) analysis of the set of transcripts down-regulated only in CSD1 metacyclics (179 transcripts) revealed statistically significant enrichment in ribosome biogenesis, rRNA processing, and ribosomal large subunit biogenesis (S1 Table), thus highlighting potential functional differences between CSD1 and CSD2.

We also analyzed the transcriptomes of the metacyclic populations for mRNAs encoding VSGs. Using 0.1% of VSG transcript as a cutoff, there were 23 VSG transcripts in RBP6 metacyclics [12,28] and 9 and 10 VSGs in the CSD1 and CSD2 metacyclics, respectively (S3 Table). All 8 known Lister 427 mVSGs were present and, as shown previously, the most abundant VSG in RBP6 metacyclics was mVSG397 representing about 50% of the VSG transcripts [12,28]. Interestingly, although mVSG397 in CSD1 and CSD2 metacyclics remained the most abundant mVSG message, its abundance increased to 77% and 73%, respectively (S3 Table).

## CSD1 and CSD2 are RNA-binding proteins

Cold shock domain containing proteins, like YB-1, often display multifunctional roles in binding DNA and/or RNA to modulate gene expression transcriptionally and post-transcriptionally. Immunofluorescence assays revealed that both CSD1 and CSD2 localized to the cytoplasm and appeared to be excluded from the nucleus in procyclic cells (S7 Fig). Polysome profiling, at the 16 h timepoint, further showed that neither CSD1 nor CSD2 associated with polysomes (S8 Fig). In addition, chromatin immunoprecipitation sequencing (CHIP-seq) with CSD1-3xFLAG or CSD2-3xFLAG overexpression at the 8 h timepoint did not reveal any evidence of DNA binding activity. To assess whether CSD1 and CSD2 interacted with RNA in

living cells, we employed single-end enhanced crosslinking and immunoprecipitation (seCLIP) followed by RNA-Seq [39] in the RBP6 overexpression cell lines that contained either CSD1 or CSD2 endogenously epitope-tagged with 3xFLAG at the C-terminus. Two seCLIP experiments on UV crosslinked biological replicate samples and two size-matched input controls (taken from each of the two UV crosslinked samples) were processed and mapped to the *T. b. brucei* TREU927 genome. To identify seCLIP peaks, we manually inspected the aligned data and considered crosslinked sites with a read count of at least fifty and at least five-fold above the input control. Although crosslink efficiency was variable between replicates, the reproducibility of crosslinks was validated by a peak of crosslink positions reproduced in an independent experiment at the same exact position. This resulted in 308 and 112 identified peaks for CSD1 and CSD2, respectively, with all the peaks called for CSD2 overlapping with CSD1 peaks (S4 Table and see S9 Fig for examples). One possibility is that CSD1 indeed binds more targets than CSD2. However, we suspect that more CSD1 targets were found, because the CSD1 data set was more complex. In particular, the peaks of CSD1 were between 5- and 10-fold higher than CSD2 (S9 Fig). In addition, when replicates were clustered by their Spearman correlations, CSD1 replicates were more similar to each other than to CSD2 replicates. Although it remains possible that CSD1 and CSD2 bind unique RNAs and that our failure to identify unique targets may be technical, we favor instead the idea that CSD1 and CSD2 bind essentially the same RNAs at the same sites (see Discussion). Henceforth, we focused our analysis on the CSD1 peaks and targets.

In order to address the RNA sequence specificity of the 308 CSD1 peaks, we used the motif analysis algorithm MEME [40] to search for sequence motifs in a 30-nt window around the RBP binding sites. This analysis retrieved ANACAU in 301 peaks as the most significantly enriched sequence motif (e-value: $1.4^{-327}$)(Fig 7B). The majority of binding sites were within coding regions (201, 67%). 85 of the binding sites (28%) were within the 3' UTR of the target transcripts and a few (15, 5%) mapped to the 5' UTR of target mRNAs. 12 transcripts had multiple peaks and thus, we identified 289 target transcripts for CSD1 (S4 Table). Functional categorization by GO analysis of the targeted transcripts revealed statistically significant biological processes, including sphingomyelin metabolism, microtubule-based movement and nucleocytoplasmic transport (Fig 7C, and S4 Table). Sphingolipid synthesis is essential for growth of *T. brucei* bloodstream forms in culture [41]. Three notable transcripts in the microtubule-based movement category encoded the katanin p80 subunit (KAT80, Tb927.9.9960), which has been implicated in reducing flagellar length and being involved in proper cytokinesis [42], an orphan kinesin localized to the ingressing furrow (KLIF, Tb927. 8.4950), which is necessary for cleavage furrow resolution [43], and trypanin (Tb927.10.6350), which is required for directional cell motility in *T. brucei* [44] and *T. cruzi* epimastigotes with a trypanin KO showed decreased efficiency of *in vitro* metacyclogenesis [45]. Speculatively, the regulation of these flagellar proteins may be involved with the shift to BSF-like motility. The transcripts encoding metacyclic VSGs were not bound by CSD1, indicating that CSD1 does not play a direct role in mVSG mRNA metabolism. Nevertheless, two target transcripts encoded essential proteins in the regulation of monoallelic VSG expression, namely the VSG-exclusion-2 (VEX2; Tb927.11.13380) [46], a protein with similarity to the nonsense-mediated-decay helicase UPF1, and TDP1 (Tb927.3.3490), a high-mobility group box protein whose overexpression disrupts VSG monoallelic expression [47].

The majority of the identified CSD1-bound mRNAs were not highlighted in our previous RNA-Seq and RNAi differential expression analyses. Nevertheless, 11 transcripts (Tb927.10.14900, Tb927.10.3620, Tb927.10.6720, Tb927.11.9590, Tb927.3.4830, Tb927.3.700, Tb927.5.1450, Tb927.5.2560, Tb927.6.1230, Tb927.7.4570, Tb927.8.7970) of the high-confidence seCLIP data set overlapped with transcripts differentially regulated during RBP6

overexpression over 6 days [28]. In addition, CSD1 depletion by RNAi in the RBP6 overexpression cell line for two days affected 167 of the up-regulated transcripts and 57 of the down-regulated transcripts [16]. Of the 57 transcripts, 34 were found to be down-regulated only in the CSD1 RNAi cell line, not in the CSD2 RNAi cell line. However, only seven transcripts (Tb927.3.4830, Tb927.3.700, Tb927.5.1450, Tb927.6.1230, Tb927.7.4570, Tb927.10.3620, Tb927.11.9590) appeared to be bound by CSD1, suggesting that a large fraction of knock-down-responsive changes in transcript abundance resulted from indirect effects. 74% of the seCLIP peaks mapped to either the 5' UTR or ORFs and there was an overlap with transcripts whose polysome association was increased or decreased at least three-fold relative to un-induced cells after 48 h of RBP6 induction [16] (13 transcripts: Tb927.10.14900, Tb927.10.3490, Tb927.10.4230, Tb927.11.14540, Tb927.11.16550, Tb927.11.2240, Tb927.11.3350, Tb927.11.8070, Tb927.3.3940, Tb927.3.4830, Tb927.5.1450, Tb927.6.1730, Tb927.8.1760). Furthermore, 7% (16 transcripts) of the CSD1 seCLIP targets were differentially regulated by at least 2-fold, when the transcript abundance was monitored in the CSD1 overexpression cell line at the 1h, 4h, 16h, and 24h timepoints. Finally, although we found that CSD1 binds predominantly to protein coding RNAs, five noncoding RNAs (ncRNAs) of unknown function were also bound [48]. In particular, the transcript with the highest peak was a ncRNA (Tb1.NT.1) of 538 nucleotides (S4 Table). The five ncRNAs are not related to each other and are unique to *T. brucei* and *T. evansi*, two closely related trypanosomatids.

## Discussion

*T. brucei* procyclic cells undergo a complex developmental process in the tsetse fly known as metacyclogenesis, which generates infectious metacyclic parasites expressing metacyclic VSGs (mVSGs). In our previously established *in vitro* differentiation system based on the overexpression of RBP6 in procyclic parasites, approximately 30% of the cells develop into epimastigotes after 2–3 days and approximately 50% of the cells are metacyclic parasites after 4–6 days [12]. In this developmental program CSD2 is required for the positioning of the kinetoplast anterior to the nucleus and thus for the appearance of epimastigote parasites [16]. In contrast, CSD1 down-regulation by RNAi did not affect the appearance of epimastigotes, but instead affected the expression of the epimastigote marker BARP [16]. In the present study we made the unexpected observation that overexpression of CSD1 or CSD2 in procyclic cells caused 40–50% of the cells to morphologically appear as metacyclics after 24 h without passing through the intermediate epimastigote stage. This rapid development to metacyclics is remarkable, especially since the transcriptome of CSD1- and CSD2-generated metacyclics is very similar to that of RBP6 metacyclics and the same 5 mVSGs emerged as the most abundant mVSGs, albeit there were differences in respective abundance levels, with mVSG397 representing about 75% of the VSG transcripts, as compared to about 50% in the RBP6 system. The skipping of the intermediate epimastigote stage was not anticipated, although we previously observed a similar phenomenon in cells expressing a RBP6 mutant, which generated metacyclics in 2–3 days [28]. It is important to point out that we have overexpressed several RBPs implicated in metacyclogenesis based on transcriptomic, proteomic or RNAi studies, and found no evidence for metacyclic production. Thus, further experiments are needed to understand how these special inducers of differentiation function.

Despite the developmental phenotypic differences observed between CSD1 and CSD2 over-expression and RNAi of CSD1 or CSD2 in the background of RBP6-driven metacyclic production, seCLIP results indicated that the binding sites of CSD1 and CSD2 are identical and that their transcript targets overlap, either partially or completely. These observations are remarkably similar to data obtained for the *Caenorhabditis elegans* PUF (Pumilio/FBF) proteins FBF-

1 and FBF-2. They are 91% identical [49], share the majority of their target mRNAs and bind to an identical binding site "UGUNNNAU" [50]. Although these proteins were initially described to be functionally redundant in maintaining germline stem cells [51], subsequent studies revealed distinct roles with FBF-1 restricting the rate of meiotic entry and FBF-2 promoting both cell division and meiotic entry [49]. Importantly, FBF-1's function depends on interactions with the CCR4-NOT deadenylase complex mediated by amino acids outside the conserved RNA binding domain [49]. FBF-2 lacks these interacting protein sequences and thus, functions independently of the deadenylase complex and seems to protect targets from deadenylation [49]. A similar scenario could explain our results with CSD1 and CSD2, where distinct functions are determined by protein sequences outside the conserved CS domain through binding to different protein partners. Indeed, besides the CSD, we found that the disordered NTD in both CSD1 and CSD2 was also essential for metacyclic production (Fig 4). There is a putative prion-like domain in the NTD, but deletion of this domain did not show any effect on metacyclic production (S3 Fig). However, the NTD in both CSD1 and CSD2 is rich in proline and serine/threonine residues. It is possible that the proline-rich sequences nucleate interactions with proteins containing the Src homology (SH) 3 (SH3) domain or other proline recognition modules, such as the WW [52] and Enabled/VASP Homology-1 (EVH1) domains [53]. Another possibility is that the highly abundant serine/threonine residues might actively regulate the function of CSD1 and CSD2 upon phosphorylation. These possibilities should be further investigated in the future.

In this study we report a comprehensive analysis of target transcripts of CSD1 in the RBP6 overexpressing cell line using seCLIP. We identified 289 target transcripts and determined that CSD1 recognizes the motif ANACAU. Many of the bound mRNAs encode nuclear, flagellar, and mitochondrial proteins that are likely involved in the morphological transition to BSF-like cells. Perhaps the most unanticipated result was that RNA abundance changes in the RBP6 overexpression cell line were seen only in a small fraction (4.7%) of the transcripts bound by CSD1 per seCLIP. In addition, changes in translation were restricted to a small fraction of the seCLIP targets (5.6%). Similar results have been reported previously and they were interpreted to indicate that "Binding by an RBP does not necessarily imply regulation of the bound target at the tested condition" [54]. Our results point to extensive parallels between the trypanosome CSD1 and CSD2 proteins and *C. elegans* FBF-1 and FBF-2, but the binding of these proteins to messages can affect different targets in different ways, challenging definitive assignment of a specific regulatory function.

## Materials and methods

### Inducible constructs, endogenous tagging of CSD1 and CSD2 and transfection

Inducible constructs of CSD1, CSD2, CSD3, and RBP16 were obtained by cloning each coding sequence into the pLew100.v5-blasticidin (BSD) resistant vector that integrates at the rRNA spacer. CSD1, CSD2, CSD3, and RBP16 coding sequences were amplified with a forward primer having the HindIII restriction site and a reverse primer containing the BamHI restriction site. The digested amplicon was ligated into the digested pLew100.v5 (BSD) plasmid and the constructs were confirmed by restriction enzyme analysis and DNA sequencing. The 3xFLAG, truncated, and domain swapped CSD1 and CSD2 constructs were all epitope-tagged at the C-terminus. Each of these constructs was obtained with the respective oligonucleotides and ligated into the same plasmid as mentioned above. The oligonucleotides used for each plasmid construction are listed in S5 Table. All constructs were linearized with NotI prior to transfection into *T. brucei* Lister 427 (29–13) strain as described [16].

Endogenous tagging of one allele of CSD1 or CSD2 with the 3xFLAG was done by targeting the tag to the CSD1 or CSD2 locus using 173 bp of the C-terminus of each coding sequence and the first 173 bp of the 3' UTR of each target. The constructs including the paraflagellar rod 2 (Tb927.8.4970) intergenic region and the puromycin resistant drug were amplified from a pXS2-triple-FLAG-Puromycin plasmid and integrated at the C-terminus of each gene by homologous recombination. The oligonucleotide used to amplify the endogenous constructs are listed in S5 Table. The constructs were transfected into the *T. brucei* Lister 427 (29–13) strain carrying the doxycycline-inducible RBP6 (Tb927.3.2930) transgene at the rDNA spacer [16]. The selective drug, puromycin (Invitrogen) was added to the culture medium 24 hours after electroporation at a final concentration of 1 μg/ml. The integration of the construct in CSD1 and CSD2 loci was confirmed by DNA sequencing.

## RNA preparation, RNA-Seq, read processing, and data analysis

Total RNA was prepared from approximately 2 x$10^7$ to 5 x$10^7$ un-induced cells, 1-, 4-, 16-, 24-hour induced cells, and purified metacyclics from induced CSD1 or CSD2 cells. The RNA was prepared using the TRIzol reagent from Invitrogen according to the manufacturer's instructions. Isolation of poly(A)$^+$ mRNA, library preparation and sequencing on an Illumina HiSeq2500 platform were performed at the Yale Center for Genome Analysis and read procession, mapping and data analysis was done as described [28]. Briefly, the reads of 75 nt were mapped to the *T. brucei* 11 megabase chromosomes (GeneDB version 5) using the Lasergene 17.1 software package from DNASTAR. The SeqMan NGen layout algorithm by DNASTAR is based on a local match percentage and the match percentage threshold has to be met in each overlapping window of 50 bases. We used a minimum aligned length of 70 nt and allowed a maximum of two mismatches. Reads with multiple matches in the genome were distributed randomly between the identical sequences, and the genes were counted as separate. For normalization, we used RPKM (reads assigned per kilobase of target per million mapped reads). For statistical analyses of differentially expressed genes (DEGs), we used DESeq2 included in the Lasergene package. Adjusted p-value (false discovery rate, FDR) was calculated using the Benjamini-Hochberg method for multi-comparison [55]. DEGs were filtered for fold-change >2 and adjusted p-value <0.05. Since VSG genes share substantial sequence similarities, read coverage was visually inspected using the DNASTAR SeqMan NGen program, and only genes with a uniform read coverage were included in the data analysis.

## Single-end enhanced crosslinking and immunoprecipitation sequencing (seCLIP)

To probe the RNA binding activity of CSD1 and CSD2, seCLIP was carried out as previously described [39] with slight modifications as described below. We used the endogenously 3xFLAG-tagged CSD1 and CSD2 in the RBP6 overexpression background. Because the development was delayed upon induction of RBP6 in the cell line containing the endogenously tagged CSD1, we combined the induced cells from day 1 to day 4, whereas 24 h induced cells were used for the RBP6-OE cell line with the endogenously tagged CSD2, since this cell line behaved as the parental cell line (RBP6-OE). Briefly, 50 μl Dynabeads-Protein G (Invitrogen, catalogue no. 10003D) was coupled with 10 μg monoclonal anti-FLAG M2 antibody (Sigma-Aldrich, catalogue no. 1804) in the seCLIP lysis buffer (50 mM Tris-HCl pH 7.4, 100 mM NaCl, 1% NP-40, 0.1% SDS, 0.5% sodium deoxycholate) at 4˚C overnight. 3.0 x $10^9$ cells were UV-cross-linked by irradiating 3 times for 300mj/cm$^2$ in the wash buffer containing 0.25 mM glucose. The cells were lysed with seCLIP lysis

buffer and RNA was partially fragmented with serial diluted RNAse I (Invitrogen, catalogue no. AM2295). The RNA binding protein-RNA complexes were captured on anti-FLAG coupled beads at 4°C for 1h. The 3' RNA linker was ligated to the dephosphorylated RNA on the washed beads overnight at 16°C. The complexes were eluted with 1.25 mg of triple-FLAG peptide (SIGMA, catalogue no. F4799) in the wash buffer (20 mM Tris-HCl pH 7.4, 10 mM MgCl$_2$, 0.2% Tween-20, 5 mM NaCl) and 10% of the eluate and 1% of the input were subjected to Western blot analysis (referred to as cold membrane) to confirm the immunoprecipitation and to determine the lengths of the protein complexes. 90% of the eluted complexes and 2% of the input were transferred to the nitrocellulose membrane (referred to as hot membrane) and the latter was cut according to the sizes of the protein complexes from the Western blot image. RNA was extracted from the membrane and concentrated using Zymo column cleanup-RNA clean & concentrator-5 (Research Products International, catalogue no. R1016) according to the manufacturer instructions. The dephosphorylated input RNA was ligated to the 3' linker overnight at 16°C followed by the purification of the input RNA with Dynabeads MyONE silane beads (Life Technologies, catalogue no. 37002D). All ligated RNAs were reverse transcribed, and the cDNA was cleaned with Dynabeads MyONE silane beads. The 5' adapter ligation was done on the beads at room temperature overnight and the ligated cDNA was purified from the beads. The libraries were amplified with indexed sequencing oligonucleotides. The CLIP libraries were denatured at 98°C for 30 s followed by 6 cycles of 98°C for 15 s, 68°C for 30 s and 72°C for 40 s; 10 cycles of 98°C for 15 s and 72°C for 60 s and a final extension at 72°C for 60 s. The input libraries were amplified using the same conditions except that the total cycles were 9 (6 + 3). DNA bands from 175 to 350 bp were excised from the agarose gel and each library was purified with NucleoSpin kit (Macherey Nagel, catalogue no. 740609) according to the manufacturer instructions. seCLIP libraries of two biological replicates of each target were sequenced at the Yale Center for Genome Analysis using an Illumina NovaSeq S4 2x100 sequencer.

Reads obtained from Illumina sequencing were processed with Galaxy (https://galaxyproject.org/citing-galaxy/) [56] prior to the alignment to the *T. brucei* reference genome. Reads were filtered using Filter by quality with the default parameters of a quality cut-off value of 20 and 90 percent of bases in sequence that must have quality equal to or higher than the cut-off value, and the 3′ ends of the sequence reads were trimmed for the Illumina adapter sequence (AGATCGGAAGAGC) using Clip adapter sequence. Reads were then collapsed to remove PCR artifacts with Collapse sequences. The unique molecular identifiers at the 5' end were trimmed with Trim sequences. Reads in FASTA format were mapped to the *T. brucei* 11 megabase chromosomes (GeneDB version 5) using the Lasergene 17.1 software package from DNASTAR.

## Miscellaneous methods

Western blot and polysome analysis were done as described [16]. The following antibodies were used: for RBP6 a rabbit antiserum raised against the peptide QPYHPFATTEPQARLYPY-HYC [12], for mVSG 397 a rabbit serum against the peptide CSDDAATYTSGSIAGTHALG [57], Dr. Isabel Roditi provided the anti-BARP antibody, and for the loading control against elongation factor 1-alpha (EF-1α, clone CBP-KK1 from Sigma- Aldrich). The purification of metacyclics is described in [13] and the chromatin immunoprecipitation sequencing (CHIP-seq) protocol is outlined in [58]. Site-directed mutagenesis was performed with the Quik-Change Lightning Site-Directed Mutagenesis Kit from Agilent according to the manufacturer's instructions. The primers used are listed in S5 Table.

## Supporting information

**S1 Fig. Overexpression of RBP16 or CSD3 in procyclic parasites.** Induced RBP16 (A) and CSD3 (B) were scored for procyclic, epimastigotes, and metacyclic parasites over a 4-day period. The various developmental stages were scored as previously described (Kolev NG, Ramey-Butler K, Cross GA, Ullu E, Tschudi C. Developmental progression to infectivity in *Trypanosoma brucei* triggered by an RNA-binding protein. Science. 2012; 338:1352–1353). Three independent biological replicates were performed and at least 100 cells were counted for each time point with means ± standard deviation (std). (C) Extracts from the induced RBP16 and CSD3 overexpression cell lines were probed for RBP6, BARP, and mVSG397 by Western blot. The expression profile was compared to that of the RBP6 overexpression cell line. (TIF)

**S2 Fig. The NTD of CSD1 and CSD2 contains a predicted prion-like domain.** We scanned CSD1 and CSD2 amino acids using the Prion-Like Amino Acid Composition (PLAAC) program [33] and the prion-like domains are highlighted with red amino acids. (TIF)

**S3 Fig. The predicted prion-like domain (PLD) in the NTD of CSD1 and CSD2 was dispensible for induction of mVSG expression.** Procyclic parasites that ectopically expressed CSD1-3xFLAG or CSD2-3xFLAG with the PLD deleted were harvested in a 2-day period for Western blot analysis. The loss of the PLD domain in both cell lines did not affect mVSG397 expression. wtCSD1 and wtCSD2 serves as the positive control. (TIF)

**S4 Fig. CSD1 and CSD2 domain swap.** (A) Illustrations of the various combinations of NTDs, non-conserved sequences in the CSDs, and CTDs that were used in the generation of chimeric CSD1 and CSD2 proteins. (B) Procyclic cells individually overexpressed one of the eight chimeric proteins. None of the eight cell lines were impaired in their ability to express mVSG397. wtCSD1 or wtCSD2 were used as positive controls. (TIF)

**S5 Fig. Conservation analysis and homology modeling of CSD1.** (A) Sequence alignment of the CSD of CSD1 with homologs from various kinetoplastid species. (B) Conservation plot of CSD1 based on the alignment in (A). (C) Ribbon diagram of YB-1 (green) in complex with its RNA target (yellow; PDB code: 5YTS). (D) Superposition of the YB-1/RNA complex structure on top of the homologous model of CSD1. Highly conserved residues in CDS1 are shown as magenta sticks. (E) Overlay of the RNA target of YB-1 on the conservation plot of CSD1 based on superposition in (D). (F) Ribbon diagram of CSD1 with all highly conserved residues highlighted in magenta. The two groups of positively charged and hydrophobic residues mutated in this study are labeled and colored in orange and cyan, respectively. (G) Electrostatic surface plot of CSD1 in the same orientation as in (F). (TIF)

**S6 Fig. Transcript abundance of epimastigote markers in RBP6, CSD1 and CSD2 overexpression cell lines.** Single cell RNA-Seq of trypanosome development in the tsetse fly vector identified 5 epimastigote markers (Vigneron *et al.* PNAS 117, 2613–2621, 2020): phosphatidic acid phosphatase alpha, putative (Tb927.10.13700), phosphatidic acid phosphatase, putative (Tb927.10.13400), SGE1 (Tb927.7.360), small kinetoplastid calpain-related protein 1–4 (Tb927.1.2150), and small kinetoplastid calpain-related protein 1–5 (Tb927.1.2160). (TIF)

**S7 Fig. CSD1 and CSD2 predominantly localize to the cytoplasm and may be excluded from the nucleus.**
(TIF)

**S8 Fig. Polysome profiling of the CSD1 and CSD2 overexpression cell lines.** Lysates obtained from un-induced (A) and 16 h induced CSD1 (B) and CSD2 (C) overexpression cell lines were loaded on a linear 15–50% sucrose gradient and the absorbance was recorded at 254 nm. The positions of 40S, 60S, 80S and polysomes are indicated. The six pooled gradient fractions are: 1–2 (free RNA), 3–4 (40S), 5–8 (60S and 80S), 9–12 (light polysomes), 13–22 (heavy polysomes). The bottom panels show Western blots performed on each designated fraction for ribosomal p0 protein or for the 3xFLAG-tagged protein.
(TIF)

**S9 Fig. Example plots of mapped peaks.** CSD1 (red line), CSD2 (blue line) and corresponding input controls (gray line) are shown. The number on the right indicates the scale of the plot area. Annotated peaks are pointed out by arrows.
(TIF)

**S1 Table. CSD1 Overexpression RNA-Seq Data.**
(XLSX)

**S2 Table. CSD2 Overexpression RNA-Seq Data.**
(XLSX)

**S3 Table. Metacyclic mVSG data.**
(XLSX)

**S4 Table. CSD1 and CSD2 seCLIP Data.**
(XLSX)

**S5 Table. List of primers used for mutagenesis and cloning.**
(XLSX)

## Acknowledgments

We acknowledge the Yale Center for Genome Analysis for Illumina sequencing. We thank Dr. Diane McMahon-Pratt for the anti-PFR antibody and Dr. Isabel Roditi for the anti-BARP antibody and Dr. Saúl Rojas Sánchez for help with the figures and the seCLIP procedure. We thank Christopher Wen and Dr. Saúl Rojas Sánchez for edits and comments of the manuscript.

## Author Contributions

**Conceptualization:** Agathe Nkouawa, Nikolay G. Kolev, Christian Tschudi.

**Formal analysis:** Justin Y. Toh, Agathe Nkouawa, Gang Dong, Christian Tschudi.

**Funding acquisition:** Christian Tschudi.

**Investigation:** Justin Y. Toh.

**Supervision:** Nikolay G. Kolev, Christian Tschudi.

**Writing – original draft:** Justin Y. Toh, Agathe Nkouawa, Christian Tschudi.

**Writing – review & editing:** Justin Y. Toh, Agathe Nkouawa, Gang Dong, Nikolay G. Kolev, Christian Tschudi.

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
