## [Decision Letter · Decision Letter 0]

7 Mar 2023

Dear Chris,

Thank you very much for submitting your manuscript "Two cold shock domain containing proteins trigger the development of infectious Trypanosoma brucei" for consideration at PLOS Pathogens. As with all papers reviewed by the journal, your manuscript was reviewed by members of the editorial board and by three independent reviewers, who were supportive of your manuscript. In light of the reviews (below this email), we would like to invite you to revise the manuscript in order to take into account the reviewers' comments and suggestions for improvement. As well as some of the presentational improvements proposed (e.g increased detail in the Figure legends to assist readers and changes to some figure clarity), you should address the methodological queries around the seCLIP analysis and normalisation to reassure on the validity of the target identification, and address the request for statistical support. Clarification on the cell type proportions used in analysis of RBP and CD1/2 perturbations would also assist with understanding of the differences in transcriptome profile observed.

We cannot make any decision about publication until we have seen the revised manuscript and your response to the reviewers' comments. Your revised manuscript is also likely to be sent to reviewers for further evaluation.

kind regards,

Keith

Keith R. Matthews

Guest Editor

PLOS Pathogens

Margaret Phillips

Section Editor

PLOS Pathogens

Kasturi Haldar

Editor-in-Chief

PLOS Pathogens

orcid.org/0000-0001-5065-158X

Michael Malim

Editor-in-Chief

PLOS Pathogens

orcid.org/0000-0002-7699-2064

Reviewer's Responses to Questions

**Part I - Summary**

Reviewer #1: Metacyclogenesis is important for the development of mammal infectious stages of Trypanosoma brucei in their insect vector. The current study reports analyses of two cold-shock domain containing proteins that can individually trigger metacyclogenesis. The authors have used over-expression, expression of truncated and mutated proteins, transciptomic and immunoprecipitation methods to characterise these putative RNA-binding proteins and their interactions with mRNA.

Reviewer #2: The authors discuss a thorough characterisation of CSD1 and CSD2 and their role in development of metacyclics T. brucei, which was identified in a previous study. This work highlights the overlapping roles of these in the metacyclic development, with some variation in phenotype when genetically manipulated, and adds to functional knowledge of CSD proteins in general. The use of mutagenesis highlights that only the N-terminal domain is needed to induce metacyclic development, and identifies several essential residues and regions. RNA-seq highlights expression of a bloodstream form-like transcriptome after overexpression of either protein, similar to RBP6 overexpression. This reinforces that these proteins are involved in metacyclogenesis, although direct comparisons to RBP6-generated metacyclics is limited as mixed population are analysed in each sample and these differ in composition (due to the assumed presence of epimastigotes in the RBP6 samples only, and not in CSD1/2 samples).

Although the limitations of seCLIP results exist, as highlighted by the authors, the approach is still likely to have identified mRNAs bound by CSD1 and CSD2, and helped resolved direct and indirect targets. However, more details and results need to be added to evaluate this.

Overall, this study is robust and supports the authors claims. Most of the following queries are requests for further information and clarity in the approaches taken.

Reviewer #3: Trypanosomes go through a series of developmental stages over the course of the life cycle. Differentiation from on stage to another is performed against a background of limited gene specific transcriptional regulation. The authors' lab made the original observation that over expression of the RNA binging protein RBP6 resulted in passage through a series of differentiations from procyclic to metacyclic forms over 5 or 6 days. They subsequently identified genes that were necessary for the RBP6-driven differentiation and this manuscript characterises two of these, CSD1 and CSD2. They find that over expression of either causes differential from procyclics to metacyclics to occur in 24 hours, a spectacular finding. They go on to show that both are RNA binding proteins and identify the transcripts and motif bound. In addition they perform a deletion analysis to show that RNA binding is necessary for induced differentiation.

This is a really good manuscript describing a thorough analysis of a very interesting phenomenon. It is erudite and a pleasure to read.

**Part II – Major Issues: Key Experiments Required for Acceptance**

Reviewer #1: Fig 2: The authors should also assess these cells using flow cytometry to determine cell cycle phase, since DAPI staining and microscopy, without further quantification, does not reveal nuclear DNA content.

Some data are difficult to assess without statistical analysis. In Fig 7B, for example, are these cohort enrichments significant (unclear what the axis represents)? It’s also unclear whether CSD1 binds the sequence shown in Fig 7C. What’s the frequency of occurrence and p-value associated with this putative ‘CSD1-binding motif’ for example when the set of 233 high-confidence targets are compared to a control set of sequences using the MEME suite?

Reviewer #2: • The difference in RNA-seq results between RBP6 and CSD1/2 overexpression are discussed. What is the composition of the population of each sample sequenced? do the differences in transcriptomes result from a lack of epimastigotes after CSD1/2 overexpression, that are present in the RBP6 overexpression sample? Are epimastigote markers unchanged in CSD samples but altered in RBP6, reflecting this?

o What is the composition of life cycle cell types in the RBP6 over expression samples when RNA was isolated? Was this performed in this study, or are comparisons made to previously published work?

o Authors claim “more pronounced BSF-like transcriptome” in CSD1/2 overexpression relative to RBP6, but is this due to the nature of the mixed populations? And less so to the nature of the resulting metacyclics themselves? Does the former contain a greater proportion of metacyclics at the shown timepoint than the latter, therefore skewing results.

• seCLIP analysis details

o what were the normalisation and peak identification steps? Was custom code used? Peaks are defined by an “arbitrary cutoff of 100” – 100 what exactly?

o Why was normalisation done to previous RNA-seq and not input controls? Shouldn’t inputs from the same sample account for RNA abundance differences?

o What exactly were these previous RNA-seq samples used for normalisation (4, 16 or 24hr time points? Combined?)? Can the authors please clarify if the resulting sample composition was similar between seCLIP and RNA-seq? Especially since CSD1 seCLIP sample is a combination of multiple time points across mixed populations. RNA abundance could be different between these experiments and so this may not be an appropriate control to normalise the peaks.

o Input samples were generated, can the authors please clarify how they were used in analysis?

o The data is not shown other than a list of peaks in the supplementary table. It would be useful to see example plots of the mapped peaks, compared to the appropriate controls. And to show difference between CSD1 and CSD2, which led to claims that peaks were likely missing in the latter and that each protein in fact has identical targets, even though these aren’t identified for CSD2.

o Sup table has 308 peaks, why were peaks some removed to leave 289?

Reviewer #3: The only major concern I have is presentational. The results sections describes the numbers of transcripts that change on over expression, the overlap with those that change on RBP6 over expression. This is a little turgid and I wonder whether the dat would be better presented in a table?

**Part III – Minor Issues: Editorial and Data Presentation Modifications**

Reviewer #1: Several legends have insufficient detail: For example, are the replicates technical or biological and what do the error bars represent in fig 1A-B? Which antibodies were used to detect the four proteins in fig 1D? Are the replicates technical or biological in Fig 2? How was Fig 7C generated?

In the abstract and author summary etc, should ‘ANACAT’ be ‘ANACAU’?

Fig 2B is cited before 2A, and Fig 3B is cited before 3A.

The authors could consider shortening the ‘transcriptome’ sections from line 254-332.

Reviewer #2: • The authors refer to “purified metacyclics” in reference to fig 2B and RNA-sew results. How were these purified? There are no methods to refer to.

• Details of RNA-seq analysis should be added. How was differential expression analysis performed? What is the significant and fold-change threshold used to select up and down regulated genes?

• Cell types in fig 1 – is this based on morphology or marker expression?

• In cell counting figures (1,S1, 2b) how many cells were counted? Example images used to score each cell type would also be useful

• Scale bar needed in microscopy Fig S6

• Figure S1 - Images to show epimastigotes would be helpful, especially as BARP was not detected by western blot.

• Fig 3B is referred to before 3A in the results text, switch for clarity?

• Lines 198-200. Authors claim all CSD mutants are expressed to similar levels, but CSD1 NTD deletions looks to be higher expressed. Although this probably does not affect the claims, as this mutant does not express mVSG. Authors could quantify the expression difference, or clarify in text.

• What is the loading control used in western blots?

• There is no figure legend for Figure S4

• VSG RNA mapping – was there a quality control step to remove multimapping reads to VSGs?

• Line 331 “most abundant message” typo? mVSG?

• Fig 7B is discussed before 7A, switch?

• GO term analysis of genes uniquely differentially expressed in metacyclics after CSD1 or CDS2 over expression may help highlight difference in function between the two.

• Legend of Fig 7b, clarify which transcripts this is a summary of (ie seCLIP or RNA-seq, CSD1/2)

• The MEME analysis revealed an interesting motif, what percentage the peaks identified by seCLIP have this motif?

o The supplementary table implies this meme analysis was performed on all 289 peaks before normalising, not just high confidence peaks as implied in text. Is this correct? Please clarify

• Lines 395-400 discuss results after RNAi depletion of CSD1, please reference these experiments.

• Please reference the polyosome experiments in lines 401-406

• Line 549 – “Filter by quality” please specify this filtering step more clearly, what was the measure of mapping quality? MapQ value?

• Line 439 of the discussion claims “seCLIP-seq results indicated that the binding site of CSD1 and CSD2, as well as their transcript targets were identical”. That is not currently supported as all seCLIP peaks were not found in both cases and this is instead assumed based on the difference in data quality, which is not shown. Could CSD2 instead really just bind a subset of CSD1 targets?

Reviewer #3: Fig 1A - difficult to distinguish between the shading for metacyclic and other

figures 4 and 5 - the figures show westerns of expression levels which are fine but do not contain what I promised in the figure title. Could the success of the constructs in causing metacyclic production be included in the figure?

line 194 - need a reference for the propensity scale

PLOS authors have the option to publish the peer review history of their article (what does this mean?). If published, this will include your full peer review and any attached files.

Reviewer #1: **Yes: **David Horn

Reviewer #2: No

Reviewer #3: No
---

## [Decision Letter · Decision Letter 1]

22 May 2023

Dear Dr. Tschudi,

We are pleased to inform you that your manuscript 'Two cold shock domain containing proteins trigger the development of infectious Trypanosoma brucei' has been provisionally accepted for publication in PLOS Pathogens. All referees recommend acceptance of the article, although there was an optional suggestion for some clarification to the text you might want to consider if appropriate.

Best regards,

Keith

Keith R. Matthews

Guest Editor

PLOS Pathogens

Margaret Phillips

Section Editor

PLOS Pathogens

Kasturi Haldar

Editor-in-Chief

PLOS Pathogens

orcid.org/0000-0001-5065-158X

Michael Malim

Editor-in-Chief

PLOS Pathogens

orcid.org/0000-0002-7699-2064

Reviewer Comments (if any, and for reference):

Reviewer's Responses to Questions

**Part I - Summary**

Reviewer #1: (No Response)

Reviewer #2: The authors have addressed points raised during review, and tell an interesting story concerning the role of CSD proteins and life cycle development. They also make use of the valuable seCLIP method. There is only one point that could still be clarified.

Reviewer #3: See previous review

**Part II – Major Issues: Key Experiments Required for Acceptance**

Reviewer #1: (No Response)

Reviewer #2: None

Reviewer #3: (No Response)

**Part III – Minor Issues: Editorial and Data Presentation Modifications**

Reviewer #1: (No Response)

Reviewer #2: The authors now explain that the seCLIP IP samples are normalised to input and they manually searched for peaks over a 50- or 5-fold.

However, it is unclear in the manuscript what the aim of further normalisation to the CDS1 overexpression uninduced RNA-seq reads was. Although the reason was outlined in the reviewer response, the impact on the results is not clear. Once the 289 target genes were found, was this done to find the most common target? Or were only some of the 289 target genes used for GO term analysis based on this new normalisation? This could be made clearer for the reader or removed if the result is not discussed anywhere.

Reviewer #3: (No Response)

PLOS authors have the option to publish the peer review history of their article (what does this mean?). If published, this will include your full peer review and any attached files.

Reviewer #1: **Yes: **David Horn

Reviewer #2: No

Reviewer #3: No

---

## [Editor Report · Acceptance letter]

31 May 2023

Dear Dr. Tschudi,

We are delighted to inform you that your manuscript, "Two cold shock domain containing proteins trigger the development of infectious Trypanosoma brucei," has been formally accepted for publication in PLOS Pathogens.

Best regards,

Kasturi Haldar

Editor-in-Chief

PLOS Pathogens

orcid.org/0000-0001-5065-158X

Michael Malim

Editor-in-Chief

PLOS Pathogens

orcid.org/0000-0002-7699-2064